# Molecular Mechanisms through Which Short-Term Cold Storage Improves the Nutritional Quality and Sensory Characteristics of Postharvest Sweet Potato Tuberous Roots: A Transcriptomic Study

**DOI:** 10.3390/foods10092079

**Published:** 2021-09-02

**Authors:** Shuqian Zhou, Lu Chen, Gang Chen, Yongxin Li, Huqing Yang

**Affiliations:** School of Food and Health, Zhejiang Agricultural & Forestry University, Wusu Street #666, Lin’an District, Hangzhou 311300, China; shuqianzhou@126.com (S.Z.); lu_chen0829@126.com (L.C.); chenggangdeng@126.com (G.C.); lei_ru@126.com (Y.L.)

**Keywords:** sweet potato, postharvest treatment, edible quality, chilling injury, transcriptome

## Abstract

Sweet potato (*Ipomoea batatas* (L.) Lam.) is a commercially relevant food crop with high demand worldwide. This species belongs to the Convolvulaceae family and is native to tropical and subtropical regions. Storage temperature and time can adversely affect tuberous roots’ quality and nutritional profile. Therefore, this study evaluates the effect of storage parameters using physicochemical and transcriptome analyses. Freshly harvested tuberous roots (Xingxiang) were stored at 13 °C (control) or 5 °C (cold storage, CS) for 21 d. The results from chilling injury (CI) evaluation demonstrated that there was no significant difference in appearance, internal color, weight, and relative conductivity between tuberous roots stored at 13 and 5 °C for 14 d and indicated that short-term CS for 14 d promoted the accumulation of sucrose, chlorogenic acid, and amino acids with no CI symptoms development. This, in turn, improved sweetness, antioxidant capacity, and nutritional value of the tuberous roots. Transcriptome analyses revealed that several key genes associated with sucrose, chlorogenic acid, and amino acid biosynthesis were upregulated during short-term CS, including sucrose synthase, sucrose phosphate synthase, phenylalanine ammonia-lyase, 4-coumarate-CoA ligase, hydroxycinnamoyl-CoA quinate hydroxycinnamoyltransferase, serine hydroxymethyltransferase, alanine aminotransferase, arogenate dehydrogenase, and prephenate dehydratase. These results indicated that storage at 5 °C for 14 d could improve the nutritional quality and palatability of sweet potato tuberous roots without compromising their freshness.

## 1. Introduction

Sweet potato (*Ipomoea batatas*) is an important crop that has recently been recognized as a functional food due to its health-promoting properties and nutraceutical components [1]. Both the leaves and tuberous roots of sweet potato are consumed, providing carbohydrates, fibers, carotenes, thiamine, riboflavin, niacin, minerals, vitamins A and C, and protein [2,3]. In 2019, the global production of sweet potato reached 91,820,929 t, with a plantation area of 7,768,870 ha (Food and Agriculture Organization of the United Nations; http://www.fao.org (accessed on 16 August 2020) [4]. China is the largest producer of sweet potatoes in the world, with an annual production of 51,992,156 t (56.6% of the world’s production) in 2019 [4].

In China, a large portion of harvested sweet potato tuberous roots is shipped for fresh consumption. Sweetness and nutritional value are fundamental quality factors for sweet potato tuberous roots. Particularly, low sweetness is an undesirable characteristic of freshly harvested tuberous roots, which negatively impacts marketability. Therefore, some sweet potato tuberous roots are routinely submitted to a postharvest sweetening process prior to their marketing. For instance, Masuda et al. [5] reported that 3–4 months of storage at 13 °C is required to sweeten “Kokei 14” sweet potato tuberous roots before meeting the sweetness requirements of the fresh market. Previous studies have demonstrated that the sugar content and sweetness of postharvest sweet potato tuberous roots could be enhanced via cold storage at 4–5 °C for more than 20 d; however, the root suffered serious CI, thus affecting its overall quality and producing off-flavors [6]. CI of chilling-sensitive produce is generally thought to be the consequence of oxidative stress caused by excessive accumulation of reactive oxygen species (ROS) under low temperature [7]. Cold-induced ROS production leads to increased electrolyte leakage and compromises membrane integrity. CI is characterized by surface pitting, dark watery patches, and internal tissue browning [8,9]. Therefore, cold storage conditions must be optimized to shorten the sweetening period while also avoiding CI.

Besides sugars, chlorogenic acid is one of the key components in sweet potato tuberous roots and an important factor leading to the quality of sweet potato tuberous roots. This functional compound possesses a wide variety of health-promoting properties, including antioxidant, antimicrobial, anti-inflammatory, and antitumor activities [10,11,12]. Therefore, monitoring its changes during the postharvest process would provide a useful indicator of the nutrient composition of the sweet potato tuberous roots.

Therefore, this study sought to develop a quick and safe method to improve the quality of freshly harvested “Xinxiang” sweet potato tuberous roots. Tuberous roots were stored at 13 °C (control) and 5 °C (cold storage, CS) for 3 weeks. Afterward, the effects of different storage temperature and time combinations on quality parameters such as mass loss, flesh color, relative conductivity, and sugar, chlorogenic acid, and amino acid contents during storage were investigated. Furthermore, transcriptomic analyses were also performed to understand the molecular mechanisms that drive the aforementioned processes, and key genes associated with quality changes were identified.

## 2. Materials and Methods

### 2.1. Plant Materials

“Xinxiang” sweet potato tuberous roots were harvested at optimum maturity (135 d) from the Lingxi sweet potato professional cooperative (Hangzhou City, China). Similarly sized tuberous roots without mechanical or biological damage were selected. The selected tuberous roots were randomly divided into two groups, each with 240 tuberous roots (3 replicates; *n* = 80). Two groups of tuberous roots were packed in non-woven bags and stored at 13 °C (control) or 5 °C (cold storage, CS), respectively, for 21 d under 80%–85% relative humidity. After 14 and 21 d, some samples were transferred and stored at 20 °C for another 3 d to simulate shelf display. Shelf display is reflected in whether there will be different changes between the exterior and interior when it is transferred to normal temperature after cold storage. Three tuberous roots were randomly selected at 0, 14 + 3, and 21 + 3 d for visual assessment of CI and flesh color. Tuberous root tissue samples from under the skin (0.5–1 cm) were collected on 0, 7, 14, and 21 d of storage frozen in liquid nitrogen and stored at −80 °C for further phytochemical and molecular analyses.

### 2.2. Relative Conductivity and Mass Loss

Relative conductivity was measured as described by Li et al. [9] with some modifications. Thirty tuberous root disks (10 mm diameter and 2 mm thickness) were put in 100 mL of deionized water and shake at 150 cycles min^−1^ for 1 h on a lab plate shaker. The solution conductivity was measured using a Model EC 215 conductivity meter (HANNA Instruments, Beijing, China). The total conductivity was obtained after each sample was boiled for 15 min and the relative electrolyte leakage was expressed as a percentage of the total conductivity. The results were calculated using the following Equation (1):Relative conductivity = (P_1_/P_0_) × 100% (1)
where P_1_ is the conductivity after vibration; P_0_ is the conductivity after boiling.

Ten marked tuberous roots were weighted, and the mass loss was calculated as follows Equation (2):Mass loss = [(W_1_ − W_2_)/W_1_] × 100% (2)
where W_1_ is the fresh mass; W_2_ is the mass after storage.

### 2.3. Sugar Content, Sweetness Index, and Taste Evaluation

The sugar content was evaluated as described by Li et al. [9] with some modifications. Briefly, 2.0 g of sample was ground in 15 mL of 40% acetonitrile in a mortar on ice. The homogenate was then transferred to a centrifuge tube and sonicated for 20 min before centrifuging at 12,000× *g* for 15 min. The precipitate was reextracted twice with 2 × 15 mL extracting solvent. All supernatants were then collected to a final 50 mL volume, after which the sample was filtered through a 0.45 µm membrane filter prior to assessment.

Next, 10 μL of the filtrate was injected into a high-performance liquid chromatography (HPLC) system (Agilent 1200, Agilent Technologies, Palo Alto, CA, USA) fitted with a refractive index detector (RID-1260, Agilent) and an Agilent ZORBAX Carbohydrate Analysis Column (4.6 mm × 250 mm, Agilent). Soluble sugars were separated in 75% (*v*/*v*) acetonitrile at a 1 mL min^−1^ flow rate at room temperature. Target peaks were identified by comparing the retention times of the compounds in the sample solutions to those of a standard mixture, and compound concentrations were determined via the area percentage method. All results were expressed as mg kg^−1^ on a dry mass basis.

The sweetness index, an estimate of total sweetness perception, is calculated based on the amount and sweetness properties of individual carbohydrates in fruits and vegetables [13]. The sweetness contribution of each carbohydrate was calculated based on the fact that fructose and sucrose are 2.30 and 1.35 times sweeter than glucose, respectively [14]. The sweetness index was calculated using the following Equation (3):Sweetness index = 1.00 × glucose + 2.30 × fructose + 1.35 × sucrose (3)

The sensory quality of sweet potato tuberous roots was evaluated by measuring the sensory quality index. Evenly sized tuberous roots were selected, cleaned, weighed, and boiled for 40 min under atmospheric pressure. After cooking, the tuberous roots were allowed to cool for 30 min prior to sensory evaluation. A group of 10 people of different ages, sex, and education levels participated in the assessment. A five-point hedonic scale: 2 = “extremely dislike”; 4 = “dislike”; 6 = “neither like nor dislike”; 8 = “like”; and 10 = “extremely like” was used to assess the sensory qualities of the boiled tuberous roots. The sensory attributes evaluated were general appearance, smell, sweetness, and overall acceptability.

### 2.4. Total Phenolics, Individual Phenolic Acid Content, and Antioxidant Capacity

Sample extracts were prepared as described by Wang et al. [15] with some modifications. Briefly, 3.0 g of sample was ground in 15 mL of 80% methanol in a mortar on ice. The homogenate was then sonicated for 20 min and centrifuged at 12,000× *g* for 15 min at 4 °C. The residue was reextracted twice with 2 × 15 mL of extracting solvent. The supernatants were then pooled, and vacuum evaporated at 40 °C to remove all methanol residues. The remaining extract was adjusted to pH 1.5 with 6 mol L ^−1^ HCl and centrifuged at 12,000× *g* for 15 min at 4 °C. Afterward, the sample was extracted five additional times with an equal volume of ethyl acetate-ether (*v*/*v*; 1:1), then pooled. The supernatant was dried with anhydrous sodium sulfate, filtered with filter paper, and vacuum evaporated at 35 °C until dry. The residue was dissolved in 5 mL methanol, and the sample was filtered through a 0.45 µm membrane prior to analysis.

A total phenolic standard curve was constructed as described by Kalt et al. [16]. The total phenolic content (TPC) of the sweet potato tuberous roots extracts were determined spectrophotometrically using Folin–Ciocalteu’s reagent, and the results were expressed as gallic acid equivalents per gram of dry mass (mg GAE kg^−1^ DW).

The levels of individual phenolic acids in tuberous roots were determined via HPLC analyses. Phenolic acid standards were prepared in methanol (*m*/*v*; 0.01:10) and stored in brown reagent bottles before use. Afterward, 10 μL of the filtrate was injected into an HPLC system fitted with a refractive index detector (RID-1260, Agilent) and an Agilent ZORBAX Eclipse Plus C18 column (4.6 mm × 250 mm, Agilent). The mobile phase was composed of (A) 2% acetic acid (aqueous) and (B) acetonitrile, and gradient elution was performed as follows: 0 min, 97:3; 20 min, 95:5; 35 min, 85:15; 65 min, 70:30; and 65 min, 0:100. The mobile phase was vacuum filtered through a 0.45 µm membrane filter before use. All HPLC analyses were conducted at a 1 mL min^−1^ flow rate at room temperature, and the results were expressed as mg kg-1 DW.

DPPH radical-scavenging activity was determined as described by Oliveira et al. [17] with some modifications. Scavenging activity was calculated using the following Equation (4):Scavenging activity = [(A_517_ of control − A_517_ of sample)/A_517_ of control] × 100(4)

DPPH scavenging activity values were expressed as mg Trolox equivalent antioxidant capacity·kg^−1^ (mg TEAC kg^−1^ DW).

Ferric reducing antioxidant power (FRAP) assays were conducted as described by Apati et al. [18] with some modifications. Briefly, 1 mL sample solution was mixed with 2.5 mL of phosphate buffer (0.2 mol L^−1^, pH 6.6) and 2.5 mL of K_3_Fe(CN)_6_ (*v*/*v*; 1:100).The mixture was incubated at 50 °C for 20 min and added 2.5 mL of trichloroacetic acid (*v*/*v*; 1:10), which was then centrifugated at 3000 rpm for 10 min. The supernatants were sucked 2.5 mL and mixed with 2.5 mL of bidistilled water and 0.5 mL of FeCl_3_ (*v*/*v*; 0.1:100). The absorbance was measured at 700 nm. The results were expressed as mg TEAC kg^−1^ DW.

### 2.5. Free Amino Acid Content Determination

A 0.5 g sample was ground in a mortar on ice with 5 mL of extracting solution containing 50% alcohol and 0.01 mol L^−1^ of HCl. The homogenate was transferred to a centrifuge tube and sonicated for 30 min at low temperature, then centrifuged at 12,000× *g* for 5 min. Next, 1 mL of supernatant was recovered, freeze-dried, and dissolved in 1 mL of amino acid diluent. The sample was filtered through a 0.22 µm membrane filter before use. Finally, 50 μL of the filtrate was assessed in an automatic amino acid analyzer (SykamS433D, SYKAM Vertriebs GmbH, Fürstenfeldbruck, Germany) with an analysis column (LCA K07/Li, Sykam). The results were expressed as mg kg^−1^ DW.

### 2.6. RNA Isolation and Sequencing

Total RNA was extracted from control samples on days 0 and 14 and from CS samples on day 14 using the RNAprep Pure Plant Kit (Tiangen Biotech Co., Ltd., Beijing, China). Agarose gel electrophoresis was used to evaluate the RNA integrity. RNA sequencing (RNA-Seq) was then conducted by BmK Biotechnology Co., Ltd. (Beijing, China). Adapter sequences and low-quality sequences were then filtered out to ensure that all downstream analyses were conducted using clean and high-quality data. The hisat2 software was then used to map the data to the wild sweet potato reference genome (http://sweetpotato.plantbiology.msu.edu/ (accessed on 15 October 2020)). The hisat2 software uses an indexing scheme based on the Burrows-Wheeler transform and the Ferragina-Manzini (FM) index, employing two types of indexes for alignment: a whole-genome FM index to anchor each alignment and numerous local FM indexes for very rapid extensions of these alignments [19]. Functional annotation, cluster analysis, protein-protein interaction network analysis, and more in-depth mining analysis were conducted thereafter.

### 2.7. Quantitative Real-Time PCR Validation

Representative differential expressed genes (DEGs) identified by RNA-Seq were selected for experimental quantitative real-time PCR (qRT-PCR) validation. Gene-specific primers were synthesized by Beijing Qingke Biotechnology Co., Ltd. (Appendix A). The RNA was reverse transcribed into complementary DNA (cDNA) using a PrimeScript™ RT reagent Kit with gDNA Eraser (Takara Bio Inc., Kusatsu (Shiga), Japan). qRT-PCR was performed using a TB Green^®^ Premix Ex Taq™ II (Tli RNaseH Plus) (Takara Bio Inc., Kusatsu (Shiga), Japan) under the following conditions: 95 °C for 30 s, followed by 40 cycles of 95 °C for 5 s and 60 °C for 30 s. Relative gene expression was calculated using the _ΔΔ_Ct method and normalized to the expression levels of *α-tubulin*. The qRT-PCR experiments were performed using at least three biological replicates, a negative control, and two technical replicates.

### 2.8. Statistical Analyses

The data were subjected to ANOVA analysis, and significant differences between means were determined using Duncan’s multiple range test at a *p* < 0.05 probability (level). All results were reported as the mean of three replicates.

## 3. Results

### 3.1. Chilling Injury, Relative Conductivity, and Mass Loss in Sweet Potato Tuberous Roots

The typical symptoms of CI in sweet potato tuberous roots are pitting, water-soaked areas, and skin depression. Moreover, fungal infestation may be observed in damaged tissues in severe cases. The tuberous root in the CS group had no CI symptoms after 14 d at 5 °C plus 3 d of shelf display, whereas slight surface pitting and internal browning developed after 21 d at 5 °C and 3 d at 20 °C. Tuberous roots stored at 13 °C did not develop CI even after 21 d of storage (Figure 1A,B).

As shown in Figure 1C, the relative conductivity of either control or CS tuberous roots increased throughout storage. No significant difference was detected between the control or CS group until day 21, where the relative conductivity was 8.5% higher than that of the control group.

Excessive mass loss can compromise the quality of fresh produce. As shown in Figure 1D, the two groups underwent substantial loss after 21 d of storage, reaching up to 11.8% and 9.3% at 13 and 5 °C, respectively. The mass loss at 13 °C was significantly higher (*p* < 0.05) than that at 5 °C.

### 3.2. Changes in Soluble Sugars, Sweetness Index, and Sensory Qualities

Soluble sugars are the main nutritional components of sweet potato tuberous roots, among which fructose, glucose, and sucrose are the main free sugars that determine their sweetness. Particularly, sucrose content was significantly higher in tuberous roots of the CS group compared with the control (*p* < 0.05) after 14 d of storage. In contrast, glucose and fructose contents did not change significantly in tuberous roots of either CS or control groups during storage (Figure 2A–C).

As shown in Figure 2D, the sweetness index in CS group tuberous roots peaked at day 14, reaching a level that was significantly higher than that of the control group. Therefore, sweet potato tuberous roots stored at 5 °C successfully achieved low-temperature sweetening. However, the sweetness index of sweet potato tuberous roots not further improved after 14 d of cold storage. Results from the sensory evaluation indicated that rates of consumer acceptance of tuberous roots did not change after 21 d of storage at 13 °C. In contrast, the sensory score of tuberous roots stored at 5 °C for 14 d was significantly higher than that of the control tuberous roots; however, these scores decreased in tuberous roots stored for 21 d at 5 °C due to flesh browning and CI-induced off-flavors after cooking (Figure 2E).

### 3.3. Changes in Total Phenolics, Free Phenolic Acid Content, and Antioxidant Capacity

The total phenol contents in tuberous roots of both groups increased after 21 d of storage (Figure 3A). Further, the total phenol content in tuberous roots of the CS group was significantly higher than that in the control group (*p* < 0.05) after 14 d of storage. Chlorogenic acid is the main phenolic compound in sweet potatoes and thus might have contributed to the increase in total phenolic. The chlorogenic acid content in CS tuberous roots was significantly higher (*p* < 0.05) than that in the control group after 14 d (Figure 3B).

Moreover, the DPPH scavenging activity in tuberous roots of the CS group was significantly (*p* < 0.05) higher than that in the control group after 7 d of storage (Figure 3C). As shown in Figure 3D, the FRAP in the CS group also showed a significant upward trend during storage, with CS tuberous roots showing significantly higher FRAP levels than the control group (*p* < 0.05).

### 3.4. Changes in Free Amino Acid Content in Sweet Potato Tuberous Roots during Storage

The free amino acids analyzed herein were glycine, phenylalanine, tyrosine, and propanine. The initial phenylalanine content was approximately 90 mg kg^−1^ FW, followed by alanine and tyrosine. The glycine content was the lowest among the four amino acids tested. Almost all of these free amino acids tended to increase continuously between two and six times during storage at temperatures (Figure 4A–D).

### 3.5. Sweet Potato Tuberous Roots Transcriptome Analysis at Different Storage Temperatures

#### 3.5.1. Data Quality Evaluation and Analysis

To gain a better understanding of the physiological changes of sweet potato tuberous roots under different storage temperatures, RNA samples were collected on days 0 (control, C0) and 14 (control and CS group, C14, and CS14). A total of 6.44 Gb to 9.51 Gb clean reads were obtained, and more than 94.0% of the reads had a Q30 quality score (Appendix A). The sequencing data were therefore deemed adequate for correlation analysis. Hisat2 was used to align the clean reads with the sweet potato reference genome (https://ipomoea-genome.org/ (accessed on 15 October 2020)), achieving a 77.64–82.57% alignment efficiency. The alignment results are shown in Appendix A.

Pearson correlation analysis was performed to validate the gene expression profiles based on the transcripts from nine different samples (Figure 5A). As expected, the results indicated that biological replicates from the same treatment group were highly correlated. Particularly, C0 and C14 were highly correlated, whereas CS14 and C0 were less correlated. The principal component analysis (PCA) (Figure 5B) results coincided with the correlation analysis findings. These findings indicated that the gene expression profile of the CS14 and C0 groups exhibited the most significant differences.

#### 3.5.2. Unigene Annotation Statistics

Unigene function annotation was conducted using the COG, GO, KEGG, KOG, Pfam, Swissprot, eggnog, and NR databases (Appendix A). A total of 24,045 single genes were annotated in eight databases, of which the NR database had the highest annotation rate, with 31,000 annotations, accounting for 99.52% of the total annotated genes.

#### 3.5.3. DEG Screening and Annotation Analysis

##### Comparison of Two DEG Groups

Degseq was used to identify differentially expressed genes (DEGs) in C0 vs. C14 and C0 vs. CS14. The screening criteria were fold change ≥2 and FDR < 0.01. Compared with the C0, 602 unigenes were upregulated, and 466 were downregulated in C14, whereas more genes were differentially expressed in the CS14 group (4850 upregulated and 5086 downregulated) (Table 1). The two groups of DEGs were annotated using the COG, eggnog, NR, Pfam, Swissprot, GO, and KEGG databases. The number of annotated genes is shown in Appendix A, and the annotation ratio of the NR database was the highest (>95%).

##### GO Database Annotation Analysis

The GO annotation system includes three main branches: cellular component, molecular function, and biological process. The two groups of DEGs were classified into 53 functional subgroups (Appendix A). As indicated in Appendix A, in terms of cellular components, the DEGs were mainly associated with seven functional sub-categories: “cell”, “membrane”, “macroscopic complex”, “organelle”, “organelle part”, “membrane part”, and “cell part”. Regarding molecular function, DEGs were mainly associated with the two functional subclasses of “catalytic activity” and “binding”. For the biological process classification, the DEGs were mainly enriched in pathways associated with “biological process”, “cellular process”, “single organization process”, “response to stimulus”, “localization”, “biological regulation”, and “cellular component organization or biogenesis”. Regarding the differences in DEG ratios between the two groups, the “cell”, “membrane-enclosed lumen”, “macroscopic complex”, “organelle”, “organelle part”, “cell part”, “structural molecular activity”, “growth and cellular component organization or biogenesis” process ratios exhibited marked differences (e.g., up to 40-fold differences). Moderately low temperatures had an important effect on the aforementioned processes at the transcriptome level in sweet potato tuberous roots stored for 14 d.

##### KEGG Functional Annotation

The enriched KEGG pathways could be divided into five categories, including “cellular processes”, “environmental information processing”, “genetic information processing”, “metabolism”, and “organic systems”. According to the KEGG pathway classification (Appendix A) and enrichment analysis (Appendix A), the DEGs were mainly associated with metabolism pathways. As shown in Appendix A, the C0 vs. C14 and C0 vs. CS14 comparisons indicated that “plant hormone signal transduction”, “protein processing in endoplasmic reticulum”, “ribosome”, “phenylpropanoid biosynthesis”, “starch and sucrose metabolism”, “biosynthesis of amino acids”, and “carbon metabolism” were the main metabolic pathways affected by different low-temperature storage conditions. Among them, the “phenylpropanoid biosynthesis”, “starch and sucrose metabolism”, and “biosynthesis of amino acids” pathways are associated with the synthesis of partial free phenolic acids, sugars, and free amino acids, respectively. The present study mainly focused on these three metabolic pathways by screening key genes linked to metabolic changes.

#### 3.5.4. Key Genes Involved in Sweet Potato Tuberous Roots Quality

As shown in Table 2, the key genes related to sucrose synthesis were sucrose synthase and sucrose phosphate synthase, among which sucrose synthase exhibited differential expression and sucrose phosphate synthase was downregulated at low temperatures. Further, upregulated genes were more abundant than downregulated genes in sucrose synthase.

The phenylalanine ammonia-lyase, 4-coumarate-coa ligase, and hydroxycinnamoyl-CoA quinate hydroxycinnamoyltransferase genes were upregulated in the phenylalanine metabolism pathway, which has been associated with chlorogenic acid accumulation in sweet potato tuberous roots at low temperatures. In the amino acid biosynthesis pathway, low-temperature storage upregulated the alanine aminotransferase, serine hydroxymethyltransferase, aromatate dehydrogenase, and aromatate dehydratase/precursor dehydratase genes. Therefore, the lower storage temperature translated to higher expression levels, which are highly associated with the accumulation of glycine, alanine, tyrosine, and phenylalanine in sweet potato tuberous roots.

### 3.6. Validation of RNA-Seq Results via qRT-PCR

The results from the transcriptome analysis were validated in a biologically independent experiment using qRT-PCR. A total of 7 DEGs linked to sweet potato tuberous roots quality were selected and analyzed, and *α-tubulin* was used as a reference gene. As illustrated in Figure 6, the relative expression levels of sucrose synthase (itf11g07860), phenylalanine ammonia-lyase (itf09g14800), 4-coumarate-CoA ligase (itf11g10280), serine hydroxymethyltransferase (itf09g04020), alanine aminotransferase (itf11g08270), arogenate dehydrogenase (itf10g08090), and arogenate dehydratase/prephenate dehydratase (itf07g12100) were all upregulated by low-temperature storage, which was consistent with our RNA-Seq results.

## 4. Discussion

Sweet potato is now globally recognized as a functional food due to its preventive and therapeutic effects against chronic diseases [1]. However, the sweetness of freshly harvested sweet potato tuberous roots is not enough, which limits the acceptance of consumers. Cold storage can improve the sweetness of sweet potato tuberous roots, but long-term storage will cause CI. The development and likelihood of CI can also depend on the sweet potato origin, variety, cultivar, harvest season, temperature, and chilling stress duration [20]. Therefore, precise control of storage temperature and time may improve tuberous root sweetness without resulting in CI. Our study demonstrated that the sweetness of mature “Xinxiang” sweet potato tuberous roots increased 2-fold after 14 d of storage at 5 °C, which significantly improved their taste according to our sensory tests. Moreover, no CI was detected at this time point. During storage at 13 °C, the soluble sugar and sweetness index of the sweet potato tuberous roots did not change significantly, which contrasted with the findings of Ru et al. [21], who demonstrated that the levels of sugars in tuberous roots stored at both 13 and 4 °C increased after 15 d. We speculate that these differences were related to the maturity of the sweet potatoes. In this study, sweet potatoes were harvested 135 d after planting, which was different from our previous study (Ru et al.) [21]. More mature tuberous roots appeared to be less sensitive to low-temperature stress [21]. Therefore, there were no observable differences in the soluble sugar contents of tuberous roots stored at 13 °C even after 21 d.

Phenolic compounds are naturally occurring chemicals that defend sweet potatoes and other plants against biotic and abiotic stressors [22,23,24]. Chlorogenic acid (5-O-caffeoyliquinic acid, CGA) is the predominant component of phenolic acids in sweet potato tuberous roots and is the primary contributor to its antioxidative, antimutagenic, and radical-scavenging properties [25,26,27]. The increase in phenolic compound contents due to low-temperature exposure may enhance the nutraceutical value of sweet potato tuberous roots. The chlorogenic acid content in “Xinxiang” sweet potato tuberous roots increased markedly after 14 d of storage at 5 °C. This is consistent with the findings of Ishiguro et al. [28], who detected a significant increase in total phenolic content in sweet potato tuberous roots after two weeks of low-temperature exposure.

Low temperature also increases the concentration of some free amino acids in sweet potato tuberous roots, thus enhancing the nutritional value of cold-stored tuberous roots. However, tyrosine is one of the main substrates of enzymatic browning. Specifically, the enzyme tyrosinase oxidizes tyrosine to produce quinones. This, in turn, leads to melanin production, resulting in browning [29]. Therefore, a large amount of tyrosine accumulated during long-term low-temperature storage can lead to browning/blackening in fruits and vegetables [30]. This is consistent with the internal browning of sweet potato tuberous roots with surface cuts after 21 d of storage at 5 °C and 3 d at room temperature.

Several studies have assessed the effects of low temperature on transcript profiles in stored sweet potato tuberous roots. For instance, Ji et al. [8] analyzed the transcriptome changes of sweet potato tuberous roots stored at an optimal (13 °C) and low temperature (4 °C) for 6 weeks and found that the tuberous roots could resist CI by inducing genes related to the biosynthesis of unsaturated fatty acids, pathogen defense, and phenylalanine metabolism. These results indicate that phenylalanine metabolism can affect the production of phenolic compounds at low temperatures, which is consistent with the results of the present study. Additionally, their study demonstrated that membrane damage may be the main cause of chilling injury, thus highlighting the critical importance of lipid metabolism to improve the stress resistance of tuberous roots under low-temperature storage conditions. Xie et al. [31] also conducted transcriptomic analyses on “xushu18” sweet potato tuberous roots and found that the expression of genes associated with carbohydrate metabolism was regulated during cold storage, leading to the accumulation of sucrose in sweet potato tuberous roots, which was consistent with the results of this study. There have been reports on long-term cold storage of sweet potato tuberous roots to study the molecular mechanism of CI in tuberous roots, but there are no transcriptome reports on short-term cold storage of sweet potato tuberous roots. In this study, RNA-Seq was used to explore the molecular mechanisms through which short-term low-temperature storage enhances sweet potato tuberous roots’ quality. The concentrations of soluble sugars, phenolic compounds, and amino acids increased significantly after short-term low-temperature storage.

Sucrose synthase (SUS; EC 2.4.1.13), sucrose phosphate synthase (SPS; EC 2.4.1.14), and acid convertase (AI; EC 3.2.1.26) are key enzymes in sucrose metabolism (Figure 7). Among them, sucrose synthase can catalyze the reversible reaction of sucrose synthesis and decomposition [32]. Previous studies have reported that low temperatures increase the activity of sucrose synthase in wheat [33]. In this study, low-temperature storage upregulated the expression of two sucrose synthase genes. SPS is a rate-limiting enzyme in the synthesis of sucrose [34]. In this study, SPS expression was downregulated in the CS treatment, suggesting that the regulation of SPS activity is highly complex and may not be affected at the transcript level [21].

In plants, CGA biosynthesis occurs downstream of the phenylpropanoid pathway (red and blue color represents upregulation and downregulation, respectively) (Figure 7). Phenylalanine generates P-coumaroyl-CoA, which in turn is catalyzed by key enzymes in the biosynthesis of chlorogenic acid, including phenylalanine ammonia-lyase (PAL; EC 4.3.1.24), cinnamic acid 4-hydroxylase (C4H; EC 1.14.14.91), 4-coumarate-CoA ligase (4CL; EC 6.2.1.12), hydroxycinnamoyl-CoA shikimate/quinate hydroxycinnamoyltransferase (HCT; EC 2.3.1.133), β-coumaroylester 3-hydroxylases (C3H; EC 1.14.14.96), and hydroxycinnamoyl-CoA quinate hydroxycinnamoyltransferase (HQT; EC 2.3.1.99) [35,36]. In this study, several key genes belonging to the chlorogenic acid synthesis pathway, such as 3 PAL, 1 4CL, and 1 HQT, were regulated by low temperature. The expression levels of PAL in the control and CS groups increased by 5.69- and 22.71-fold, 4CL increased by 1.17- and 2.72-fold, respectively, and HQT increased by 2.20- and 4.47-fold, respectively. Further, some key genes in the shikimic acid pathway were also upregulated by low temperature, which might have contributed to the accumulation of tyrosine in CS tuberous roots.

## 5. Conclusions

Our study demonstrated that the sweetness index, antioxidant activity, amino acid content, and sensory characteristics of postharvest mature sweet potato tuberous roots were significantly improved after 14 d of storage at 5 °C without any indications of CI, but a continued extension of CS time would lead to cold injury, resulting in the decline of storage tolerance. The key genes associated with sweet potato quality parameters were also identified through RNA-Seq and verified using qPCR, and exhibited the involvement of sucrose synthase and sucrose phosphate synthase in starch and sucrose metabolism pathway, phenylalanine ammonia-lyase, 4-coumarate-CoA ligase, and hydroxycinnamoyl-CoA quinate hydroxycinnamoyltransferase in phenylpropane biosynthesis pathway and serine hydroxymethyltransferase, alanine aminotransferase, arogenate dehydrogenase and prephenate dehydratase in the biosynthesis of amino acids pathway, thus providing insights into the molecular mechanisms by which short-term cold storage enhances sweet potato quality and nutritional profile. Therefore, short-term CS treatment is an effective method to improve the nutritional and sensory quality of sweet potato tuberous roots.

## Figures and Tables

**Figure 1 foods-10-02079-f001:**
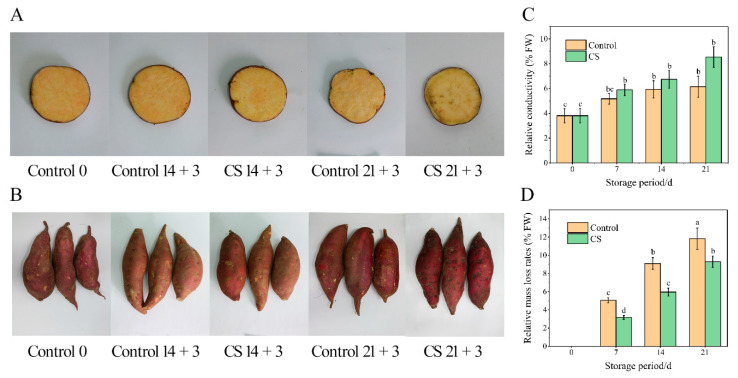
The color changes, relative conductivity, and mass loss of sweet potato tuberous roots during 21 d storage at the control and CS group. (**A**) color changes on the cutting surface, (**B**) color changes on the skin, (**C**) relative conductivity, (**D**) relative mass-loss rates. Results are expressed as the mean ± standard error of three replicates. Different characters indicate significant differences between treatment means at the *p* < 0.05 level.

**Figure 2 foods-10-02079-f002:**
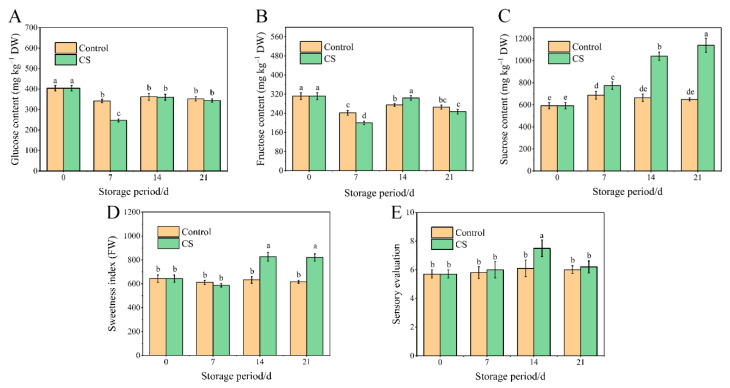
Contents of (**A**) glucose, (**B**) fructose, (**C**) sucrose, and (**D**) sweetness index and sensory score (**E**) of sweet potato tuberous roots of control and CS group during 21 d storage. Results are expressed as the mean ± standard error of three replicates. Data with different characters are significantly different from each other at the *p* < 0.05 level.

**Figure 3 foods-10-02079-f003:**
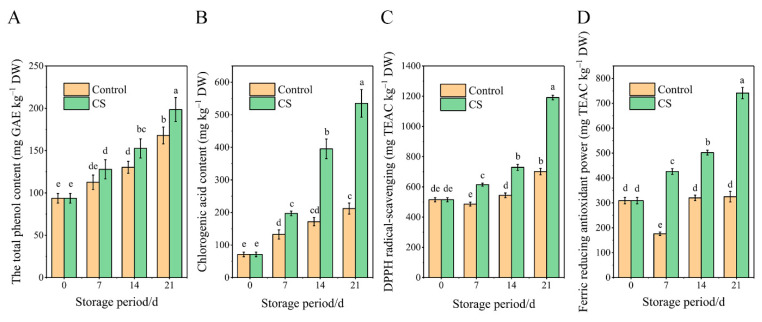
(**A**) Total phenol and (**B**) chlorogenic acid contents, (**C**) 2,2-Diphenyl-1-Picrylhydrazyl (DPPH) radical-scavenging activity, and (**D**) ferric reducing antioxidant power (FRAP) of sweet potato tuberous roots during 21 d storage control and CS groups. Data are the mean ± standard error of three replicates. Data with different characters indicate significant differences between treatments at the *p* < 0.05 level.

**Figure 4 foods-10-02079-f004:**
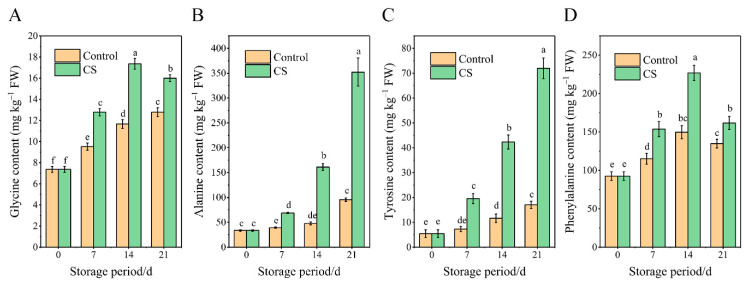
Contents of (**A**) glycine, (**B**) alanine, (**C**) tyrosine, and (**D**) phenylalanine of sweet potato tuberous roots of control and CS group during 21 d storage. Results are expressed as the mean ± standard error of three replicates. Different characters indicate significant differences between means of treatments at the *p* < 0.05 level.

**Figure 5 foods-10-02079-f005:**
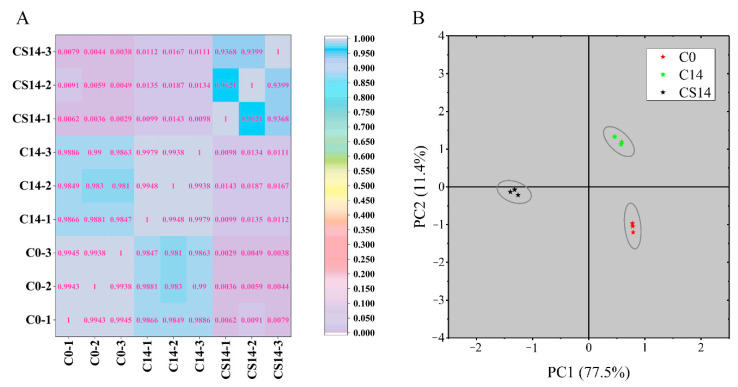
Global transcriptomic changes at the control and CS groups after 14 d of storage (C14 and CS14). (**A**) 2D hierarchical clustering, blue indicates a high correlation, purple represents low correlation. (**B**) Principal component analysis (PCA) of the RNA sequencing data of sweet potato tuberous roots samples.

**Figure 6 foods-10-02079-f006:**
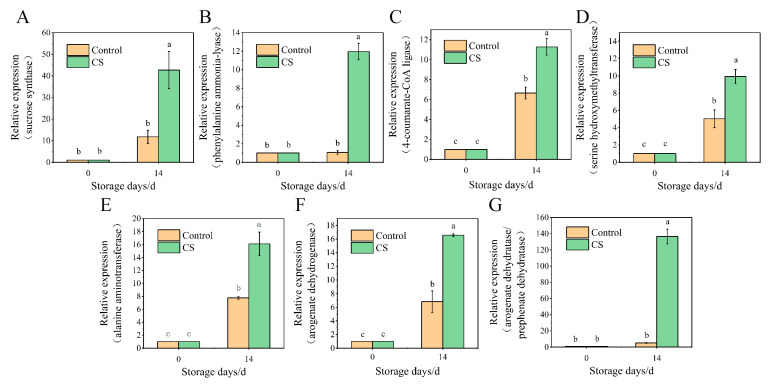
Validation of differentially expressed genes (DEGs) by quantitative real-time PCR. All data were assessed by ANOVA, and the results are expressed as the mean ± standard error of three replicates. Data with different characters indicate significant differences between treatments (*p* < 0.05).

**Figure 7 foods-10-02079-f007:**
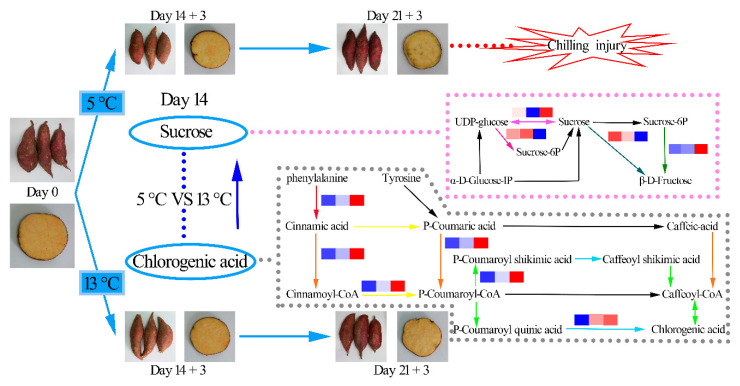
Color, sucrose content, and chlorogenic acid content changes in sweet potato tuberous roots during 14 d of storage. The figure contains the metabolite map of the sucrose and chlorogenic acid synthesis pathway. The relative expression levels of C0, C14, and CS14 are shown as heat maps.

**Table 1 foods-10-02079-t001:** Statistics of the number of deferentially expressed genes.

DEG Set	DEG NUMBER	Upregulated	Downregulated
C0 vs. C14	1068	602	466
C0 vs. CS14	9936	4850	5086

**Table 2 foods-10-02079-t002:** Selected genes associated with the accumulation of sucrose, amino acid, and chlorogenic acid in the biosynthesis pathways.

Name	Gene ID	FPKM			Enzyme
		C0	C14	CS14	
sucrose synthase	itf11g07860	88.50	67.61	912.29	EC:2.4.1.13
	itf06g18950	4.20	7.05	113.25	EC:2.4.1.13
	itf02g07130	555.91	451.65	72.19	EC:2.4.1.13
sucrose phosphate synthase	itf03g21140	84.66	96.66	19.06	EC:2.4.1.14
phenylalanine ammonia-lyase	itf09g14800	10.94	51.55	231.74	EC:4.3.1.24
	itf09g14820	6.15	30.92	138.57	EC:4.3.1.24
	itf15g00190	1.69	24.46	56.15	EC:4.3.1.24
4-coumarate-CoA ligase	itf11g10280	61.32	70.53	166.77	EC:6.2.1.12
hydroxycinnamoyl-CoA quinate hydroxycinnamoy- ltransferase	itf07g23450	100.19	220.02	448.15	EC:2.3.1.133
serine hydroxymethyltransferase	itf09g04020	26.86	32.19	60.77	EC:2.1.2.1
alanine aminotransferase	itf11g08270	44.75	47.50	283.38	EC:2.6.1.2
arogenate dehydrogenase	itf10g08090	3.86	5.57	33.58	EC:1.3.1.78
	itf12g23730	1.77	2.48	19.53	EC:1.3.1.78
arogenate dehydratase/ prephenate dehydratase	itf07g12100	0.16	3.86	12.61	EC:4.2.1.91 4.2.1.51
	itf14g14710	11.57	14.87	27.00	EC:4.2.1.91 4.2.1.51
tyrosine aminotransferase	itf02g15070	38.52	20.45	7.71	EC:2.6.1.5

## Data Availability

The data presented in this study are available on request from the corresponding author.

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
