# Peer review of "Molecular Mechanisms through Which Short-Term Cold Storage Improves the Nutritional Quality and Sensory Characteristics of Postharvest Sweet Potato Tuberous Roots: A Transcriptomic Study"

_foods, 2021, doi:10.3390/foods10092079_

Round 1

Reviewer 1 Report

In their manuscript, the authors evaluated the potential chilling effects of cold storage (5 °C vs. 13 °C; for up to 21 d) on freshly harvested sweet potato roots by analysing the nutritional and the sensory quality. They also provided a detailed transcriptome analysis of several key genes associated with sucrose, chlorogenic acid, and amino acid biosynthesis. Thus, in total, they presented many interesting results.

The topics certainly fits the scope of Foods. The experiments seem well planned, performed and evaluated. In general, the manuscript is written adequately, though there are quite a lot of mostly editorial and formal problems as marked directly in the pdf-file of the paper.

Furthermore, the discussion section could be improved. It now nearly exclusively consists of “others confirm own findings”, while a real comprehensive discussion and a convincing synopsis of the overall findings is missing. In addition, too many results are simply repeated and not summarised informatively. In this context, general molecular and genetic background and the mechanisms of low temperature sweetening was studied on potatoes in detail decades ago. It might be useful and informative, if the authors would refer to these studies. The conclusion seems very short.

From this all, I can recommend this manuscript need a moderate revision.

Some general comments:

Keywords: As electronic search engines focus on both Title and Keywords, it is better not to repeat words already used in the title. This increases the chance to get listed.

Please always replace the term “weight” by “mass”. Although still very often used instead of mass, not only in everyday language, physically correctly weight denotes a force, given in N, which is derived from mass (Fw = m * g). See relevant books of physics or engineering, or, simply, http://en.wikipedia.org/wiki/Mass.

SI requires that numerals be followed by proper SI units, e.g., 21 d, not 21 days (min for minute(s), h for hour(s), s for second(s)). Likewise, words should not be followed by unit abbreviations, e.g., twelve days, not twelve d.

SI does not permit the use of intervening or modifying words among the terms in units; e.g., FM for fresh mass is not permitted, or mg / 100 g. Explain when first needed in your materials and methods section that results are expressed on a fresh mass basis, and use for example g kg-1 or mol kg-1 in the remainder of your manuscript. You may also add the descriptor ‘FM’ after the unit.

Please, consistently either use the form mL/min or mL min-1. Better use the capital letter “L” instead of the small letter “l”.

Please write scientific species names in italics.

Reviewer 2 Report

Zhou et al. presented their findings concerning the effects of short-term storage at chilling temperatures on the quality of sweet potatoes. The article is suitable for Foods but need a minor revision, especially of the introduction which can provide more information about the scope of the research and of the discussion where the effects on the physiology of the product should be elaborated more.

My detailed comments follow below. 

Line 45 revise “consumption” to “marketability”

Line 54 this paragraph is a bit unconnected, please improve cohesion

Line 76 elaborate “shelf display”

Line 81 provide more information about the method

Line 89 I believe that W2 does not refer to dry weight but to weight after storage

Line 150 provide more information about the method

Line 210 revise “sugar”

Figure 6 replace the gene number with the gene name

Lines 363-376 this part is more suitable for the introduction

Line 386 provide reference

Line 460 the reduced storability of the product stored at lower temperatures after the treatments, should be mentioned.
